# Design Optimization of Selective Lithium Leaching of Cathodic Active Materials from Spent Lithium-Ion Batteries Based on the Taguchi Method

Yeon Jae Jung [1,2], Bong Young Yoo [2], Sung Cheol Park [1] and Seong Ho Son [1,*]

1   Advanced Functional Technology R&D Department, Korea Institute of Industrial Technology, Incheon 21999, Korea; yeonjae89@kitech.re.kr (Y.J.J.); schpark@kitech.re.kr (S.C.P.)
2   Department of Materials Science and Chemical Engineering, Hanyang University, Ansan 15588, Korea; byyoo@hanyang.ac.kr
*   Correspondence: shson@kitech.re.kr; Tel.: +82-032-850-0242

**Abstract:** The use of lithium-ion batteries (LIBs) has increased in recent years. Thus, efficient recycling is important. In this study, the Taguchi method was used to find the optimal selective lithium leaching parameters for spent LIB recycling. Orthogonal array, signal-to-noise ratio, and analysis of variance were employed to investigate the optimization of selective lithium leaching. The experimental parameters were heat treatment and leaching conditions. The lithium leaching ratio was analyzed by inductively coupled plasma (ICP). The reaction temperature was analyzed by thermogravimetry differential scanning calorimetry (TG-DSC) using lithium cobalt oxide (LCO) and carbon powder, and X-ray diffraction (XRD) was performed after heat treatment at different temperatures. From the XRD analysis, a $Li_2CO_3$ peak was observed at 700 °C. After heat treatment at 850 °C, a peak of $Li_2O$ was confirmed as $Li_2CO_3$ decomposed into $Li_2O$ and $CO_2$ over 723 °C. The $Li_2O$ reacts with $Co_3O_4$ at a high temperature to form LCO. The phase of lithium in the LIB changes according to the conditional heat treatment, affecting the lithium leaching rates. As heat treatment conditions, $N_2$ atmosphere combined with 700 °C heat treatment is suitable, and the solid–liquid ratio is important as a leaching factor for selective lithium leaching.

**Keywords:** Taguchi method; optimization; carbothermic reaction; reduction; selective leaching; lithium carbonate

## 1. Introduction

Recently, the use of lithium-ion batteries (LIBs) has increased sharply owing to the popularization of electric vehicles (EVs) and portable devices. In particular, in the case of EVs, usage is increasingly rapid, and unlike portable devices, large-capacity batteries are used [1]. According to the International Energy Agency (IEA), the usage of EVs is expected to increase rapidly. Furthermore, EV-related policies and total cost of ownership (TOC) savings are expected to greatly increase the EV market size. The number of electric vehicles on the road is estimated to reach 220 million by the 2030s [2,3].

In the case of EVs and portable devices, lithium-ion secondary batteries, which have the characteristics of high energy density and low weight, are mainly used [4,5]. Therefore, research on the treatment and re-materialization of spent LIBs has become crucial amid the increase in battery market size.

Materialization technology for recycling of spent batteries can mitigate material shortages and price increases, further leading to cost savings of major materials. Recycling technology can reduce the cost of cathode materials (NCM: nickel, cobalt, manganese) from USD 25/kg to USD 10/kg. In general, the recycling of cobalt (Co), nickel (Ni), and manganese (Mn) found in LIBs is actively carried out [6]. On the other hand, Li, which accounts for about 4–7% of battery content, has not been recovered for economic reasons, but research on Li recovery is now underway because of the recent increase in Li prices [7–9].

In the case of existing processes, one drawback is that the removal of impurities is difficult when Li is recovered from black powder because Li is extracted after recovering other valuable metals (Co, Ni, Mn, etc.). Because of this problem, research on selective Li leaching technology has been conducted, and its overview information is shown in Figure 1 [10].

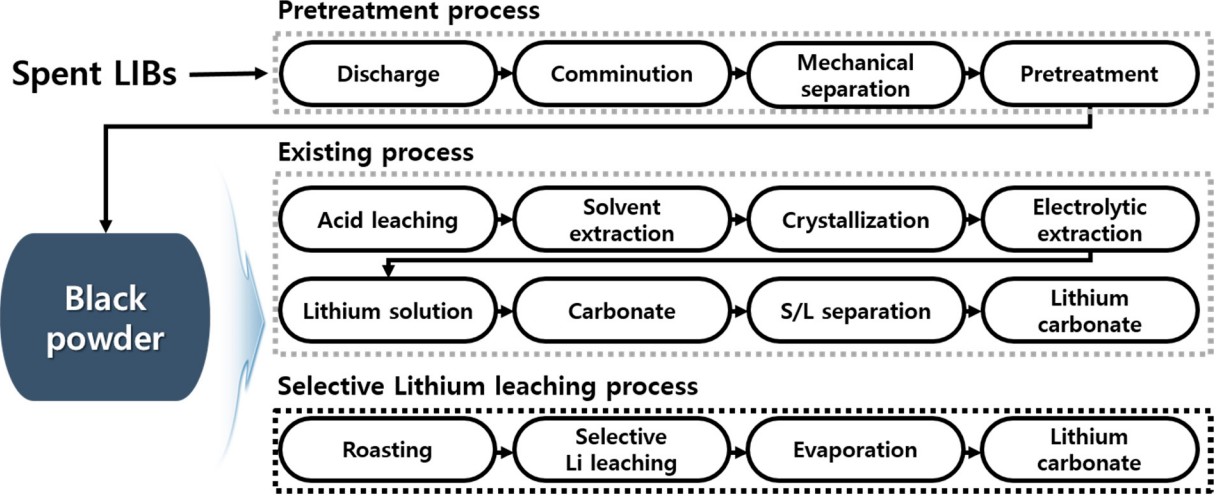

**Figure 1.** Schematic diagram of existing process and selective lithium leaching process.

In selective Li leaching, roasting is performed to convert Li compounds and allow Li leaching in water. After leaching Li, vacuum evaporation or entraining gas evaporation is performed to facilitate the recovery of Li compounds. This requires a chemical reaction in which a portion of Li in the active material ($LiCoO_2$ or LCO) reacts with C and changes to $Li_2CO_3$ [11,12].

$$4LiCoO_2 + 2C \rightarrow 2Li_2CO_3 + 4Co + O_2 \tag{1}$$

$$2LiCoO_2 + 2C \rightarrow Li_2CO_3 + 2Co + CO_{(g)} \tag{2}$$

$$4LiCoO_2 + 3C \rightarrow 2Li_2CO_3 + 4Co + CO_{2(g)} \tag{3}$$

This has the implication that when an organic material containing C exists, $LiNi_xCo_yMn_zO_2$ can be reduced to $Li_2CO_3$, Ni, Co, and MnO by the carbon reduction reaction [10,13]. After roasting, $Li_2CO_3$ can be selectively leached in deionized (DI) water. However, $Li_2CO_3$ is known to have very low solubility (12.9 g/L, 25 °C) [14]. Therefore, it is necessary to study the optimal roasting conditions and selection of the liquid-to-solid (L/S) ratio.

For effective Li leaching and recovery, this study conducted an experimental design based on the Taguchi method to analyze the effects of selective Li leaching and investigate the optimal conditions. Through these studies, it is possible to establish optimal conditions for the selective lithium leaching.

## 2. Description of the Taguchi Method

In 1920, statistician Ronald A. Fisher simultaneously studied the effects of several variables and developed a powerful statistical technique, design of experiments (DOE) [15]. It is a complete factorial design that identifies all combinations for the given factors [16]. However, a complete factorial design in the experiments of the new industry requires too many experiments. Therefore, a partial factorial design can be performed to select only a small set in all experimental combinations and reduce the number of experiments.

In this study, an orthogonal array (OA) is used based on the Taguchi approach. The Taguchi approach determines factors, levels, and analysis methods. The experimental results at this stage are contrasted with those of the conventional techniques. The Taguchi experimental design has consistency and reproducibility that are rarely found in other

statistical methods. It can be used to obtain optimal values with minimal experiments. Furthermore, it can determine a factor that has the greatest impact on the characteristics applied in the DOE. Analysis of variance (ANOVA) uses the sum of squares, degree of freedom, variance, and variance ratio to calculate the quantity, and it composes them in a standard table format. Statistically significant factors were determined within the confidence interval verified through ANOVA [16]. The DOE using the Taguchi method in experiment 3.2 was performed with two individual values (levels) in the case of heat treatment atmosphere variables (factors), and three individual values (levels) in the case of four variables (powder size, roasting temperature and time, and L/S ratio). Table 1 shows the heat treatment atmosphere, temperature, powder size, and leaching L/S ratio.

**Table 1.** Selective lithium leaching parameters and their levels.

| Symbol | Selective Li Leaching Parameter | Unit | Level 1 | Level 2 | Level 3 |
|---|---|---|---|---|---|
| A | Atmosphere | L/min | 0.5 (Air) | 0.5 ($N_2$) | - |
| B | Powder size | μm | 0–25 | 25–125 | Over 125 |
| C | Roasting Temp. | °C | 550 | 700 | 850 |
| D | Roasting time | Hour | 1 | 2 | 3 |
| E | Solid-liquid ratio | g:ml | 1:20 | 1:25 | 1:30 |

The following steps are included in the parameter design of the Taguchi method: (1) selection of characteristic evaluation targets and design parameters to be evaluated; (2) determination of the number of parameters and interactions between the parameters; (3) selection of appropriate OA and allocation of parameters; (4) stage of performing experiments based on OA; (5) analysis of experimental results using signal-to-noise (S/N) analysis and ANOVA; (6) selection of the optimal design parameter level and verification through experiments.

Three goals can be achieved through the parameter design of the Taguchi method: (1) determine the optimal design parameters for a process or product; (2) estimate each design parameter for the contribution of product quality characteristics; and (3) predict the product quality characteristics based on the optimal design parameters.

## 3. Materials and Methods

The experiments focused on selective Li first leaching from pretreated spent LIB powder. The experiments were conducted according to the process illustrated in Figure 2 by applying the OA $L_{18}$ ($2 \times 3^4$) to the heat treatment conditions and L/S ratio.

For selective Li leaching in DI water, phase separation through the reduction of $LiNi_xCo_yMn_zO_2$ and $LiCoO_2$ is crucial. The oxides containing Li react with C at high temperatures to form $Li_2CO_3$, and leaching can be performed in DI water. Therefore, the effects of heat treatment atmosphere, temperature, and time on the selective Li leaching were analyzed through X-ray diffraction (XRD) analysis.

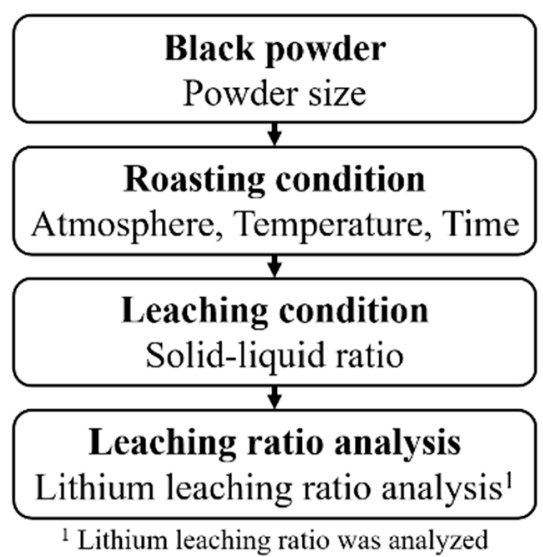

**Figure 2.** Selective lithium leaching experimental process.

### 3.1. Materials and Characterization

The black powder used in the experiments was made from LIBs that were crushed and ground after physiochemical separation at a domestic battery recycling company in Korea. Spent cell powders of EVs contain different metal components and contents based on their characteristics. Therefore, analysis was performed using XRD (X'Pert-pro mPD, PANalytical, Malvern, UK) to confirm the phase of the powder, and the content of several metals was analyzed through inductively coupled plasma optical emission spectrometer (ICP-OES, Integra XL, GBC Scientific, Melbourne, Victoria, Australia) analysis after leaching the powder in aqua regia. As shown in Figure 3, the analysis results showed that carbon and oxides containing Li, such as NCM and LCO, were present. Table 2 shows the content of each metal based on the results of ICP analysis after leaching the pulverized powder in the aqua regia. The carbon content is known to account for 35% of cell powder [17]. While performing TGA analysis in the nitrogen and air atmospheres to select the effects of temperature in the experiments, it was observed that oxygen and the organic material containing carbon react at 437 °C or higher, thereby showing a decrease in weight, as shown in Figure 4. In the air atmosphere, the weight decreased up to 750 °C and no further decrease was observed at higher temperatures. This is determined to be due to removal by the reaction of organic material containing C and oxygen in the atmosphere. On the other hand, when the experiment is conducted in a nitrogen atmosphere, the decrease in weight becomes significantly large starting at 723 °C. Additionally, compared to the experiment in the air atmosphere, the weight decreased more overall in a nitrogen atmosphere.

**Table 2.** Spent lithium-ion battery (LIB) powder ICP analysis result.

| ICP Analysis | Component (Wt.%) | | | | | |
|---|---|---|---|---|---|---|
| | Ni | Co | Mn | Li | Al | Fe |
| Black powder | 2.07 | 32.91 | 1.6 | 4.52 | 3.34 | 8.23 |

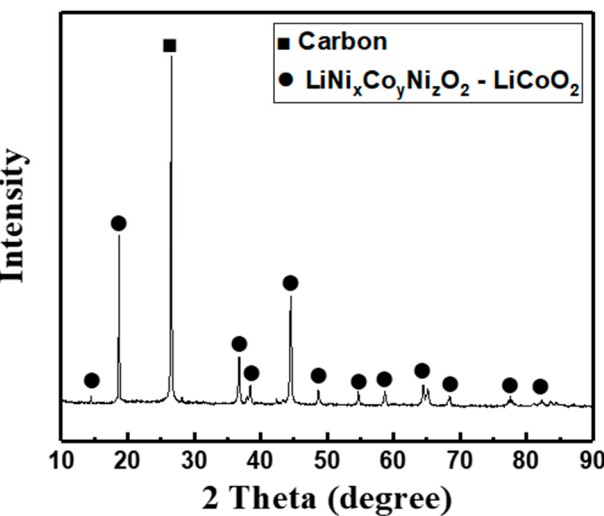

**Figure 3.** Spent LIB powder XRD analysis.

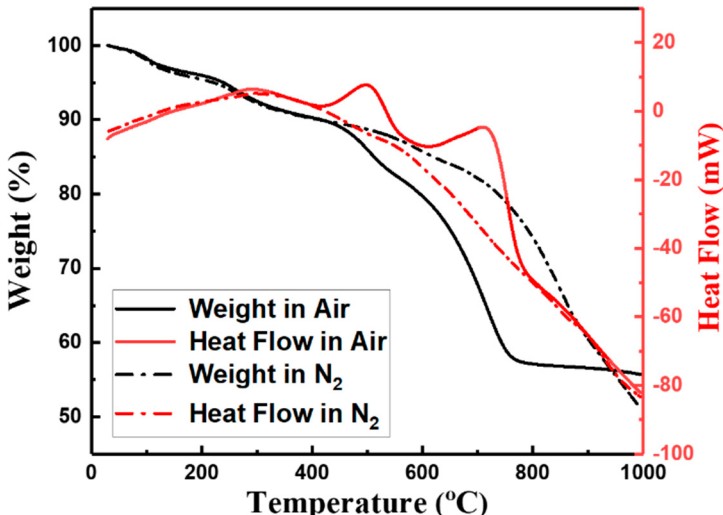

**Figure 4.** Thermogravimetric analysis in nitrogen and air atmospheres.

An XRD analysis was performed to examine the phase changes by temperature, and the results are shown in Figure 5. As shown in Figure 5a, when the heat treatment is performed in an air atmosphere, the reduction of oxides containing Li does not occur smoothly at 550 °C, and the occurrence of $Li_2CO_3$ is observed at 700 °C. However, when the heat treatment is performed at 850 °C, there is no carbon peak, indicating that reaction with the oxygen in air had occurred and no additional reduction had occurred. Furthermore, the peaks of $Li_2CO_3$ disappeared and the peaks of $LiNi_xCo_yMn_zO_2$ and $LiCoO_2$ were observed.

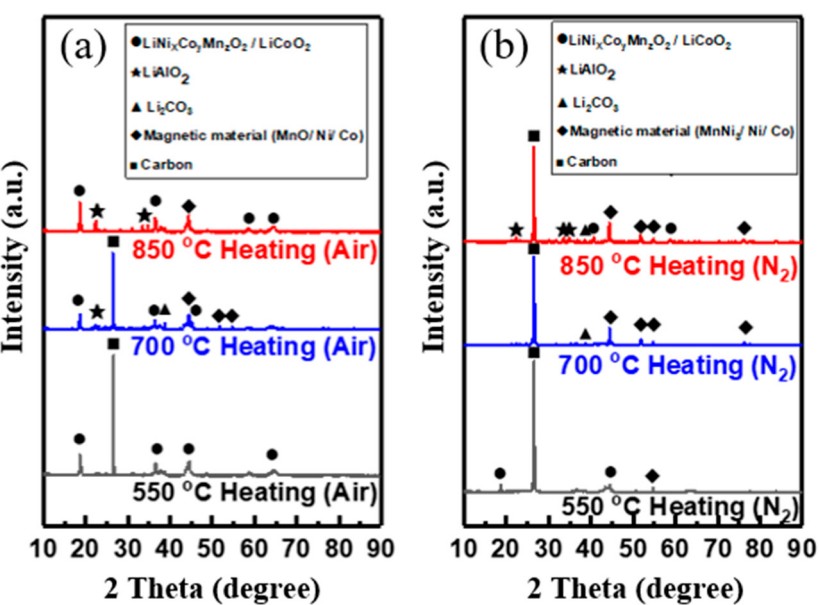

**Figure 5.** XRD analysis (**a**) in air atmosphere and (**b**) in nitrogen.

When the heat treatment is performed in nitrogen atmosphere, peaks of Co, Ni, etc. are shown because of the reduction of oxides containing Li, as shown in Figure 5b.

$$Li_2CO_3 + Al_2O_3 \rightarrow LiAlO_2 + CO_2 \tag{4}$$

A peak of $Li_2CO_3$ is observed at 700, 850 but since the peak of $LiAlO_2$ is observed at 850 °C, it is determined that the Li compounds react with $Al_2O_3$ at a temperature of 700 °C or above [18].

### 3.2. Selecting the Selective Lithium Leaching Parameters and Their Levels

Using the Taguchi method, the experiment was conducted 18 times with a total of four factors: heat treatment atmosphere and temperature, powder grain size, heat treatment time, and L/S ratio. The OA table $L_{18}$ ($2 \times 3^4$) was used by designing the heat treatment atmosphere of air and $N_2$; the powder grain size in three levels of 0–25, 25–125, and 125 μm or higher; the heat treatment temperature in three levels of 550, 700, and 850 °C; the heat treatment time in three levels of 1, 2, and 3 h; L/S ratio in three levels of 1:20, 1:25, and 1:30. Then, analysis was performed for S/N ratio, ANOVA, and interaction.

### 3.3. Lithium Leaching Ratio Measure

After performing the heat treatment for selective Li leaching, leaching was performed in deionized water. Then, after separating the liquid and the solid using glass filter (GF/C), liquid analysis was performed using ICP to analyze the amount of leached Li and calculate the Li leaching ratio.

## 4. Design and Analysis of the Leaching Parameters

### 4.1. Orthogonal Array Experiment

The total degree of freedom must be calculated to select the appropriate OA for the experiment. The degree of freedom is defined as the number of comparisons between the design parameters required to determine which level is superior. In this study, there are 17 degrees of freedom since there are five parameters for selective Li leaching, while there are two levels for the heat treatment and three levels for the remaining four parameters. If the required degree of freedom is known, the appropriate OA should be selected according to the specific task in the next step. In this study, the $L_{18}$ OA with 18 rows was used.

When the OA of $L_{18}$ was used, the effects of five parameters can be studied through 18 experiments. The experiment was performed for five parameters of selective Li leaching, as shown in Table 3.

**Table 3.** Experimental layout using an $L_{18}$ orthogonal array.

| Experiment No. | Selective Lithium Leaching Parameter Level | | | | |
|---|---|---|---|---|---|
| | **A** | **B** | **C** | **D** | **E** |
| | **Atomosphere** | **Powder Size** | **Roasting Temp.** | **Roasting Time** | **Solid-Liquid Ratio** |
| 1 | 1 | 1 | 1 | 1 | 1 |
| 2 | 1 | 1 | 2 | 2 | 2 |
| 3 | 1 | 1 | 3 | 3 | 3 |
| 4 | 1 | 2 | 1 | 1 | 2 |
| 5 | 1 | 2 | 2 | 2 | 3 |
| 6 | 1 | 2 | 3 | 3 | 1 |
| 7 | 1 | 3 | 1 | 2 | 1 |
| 8 | 1 | 3 | 2 | 3 | 2 |
| 9 | 1 | 3 | 3 | 1 | 3 |
| 10 | 2 | 1 | 1 | 3 | 3 |
| 11 | 2 | 1 | 2 | 1 | 1 |
| 12 | 2 | 1 | 3 | 2 | 2 |
| 13 | 2 | 2 | 1 | 2 | 3 |
| 14 | 2 | 2 | 2 | 3 | 1 |
| 15 | 2 | 2 | 3 | 1 | 2 |
| 16 | 2 | 3 | 1 | 3 | 2 |
| 17 | 2 | 3 | 2 | 1 | 3 |
| 18 | 2 | 3 | 3 | 2 | 1 |

### 4.2. Analysis of the S/N Ratio

In the Taguchi method, the term "signal" denotes the desired value (mean) for the output characteristics, and "noise" denotes the value (S.D.) that interferes with the output object. Therefore, the S/N ratio is the ratio of the mean to S.D. The Taguchi method uses the S/N ratio to analyze the product quality characteristics that deviate from the desired values. The S/N ratio is classified into three categories of the nominal-the-best characteristic, the smaller-the-better characteristic, and the larger-the-better characteristic. In this study, the larger-the-better characteristic is used because the greater the characteristic ($h$) for Li leaching, the better the result.

$$h = -10 \log \left( \frac{1}{n} \sum_{i=1}^{n} \frac{1}{y_i^2} \right) \tag{5}$$

In Equation (5), n is the number of tests and $y_i$ is the value of the $i$th Li leaching. As the S/N ratio increased, the deviation for the experiment decreased based on the desired value (with a high Li leaching ratio).

Table 4 shows the results of selective Li leaching. Because the DOE has a larger-the-better characteristic defined by Equation (5), the effects of each Li leaching can be separated. When the S/N ratio mean is calculated for the experiments, the results can be obtained, as shown in Table 5, and because of the larger-the-better characteristic, as the mean of S/N increases, the impact on the Li leaching increases. Furthermore, as the deviation of the mean S/N increases, the impact on Li leaching increases. Therefore, the impacts are in the descending order of heat treatment atmosphere, heat treatment temperature, heat treatment time, powder size, and L/S ratio.

**Table 4.** Experimental results for lithium leaching and signal-to-noise (S/N) ratio.

| Experiment No. | Atomosphere | Powder Size (μm) | Roasting Temp. (°C) | Roasting Time (h) | Solid-Liquidratio (g:mL) | Li Leaching Ratio (Wt.%) | S/N Ratio (dB) |
|---|---|---|---|---|---|---|---|
| 1 | Air | 0–25 | 550 | 1 | 1:20 | 25.17 | 28.02 |
| 2 | Air | 0–25 | 700 | 2 | 1:25 | 29.58 | 29.42 |
| 3 | Air | 0–25 | 850 | 3 | 1:30 | 11.16 | 20.95 |
| 4 | Air | 25–125 | 550 | 1 | 1:25 | 16.26 | 24.22 |
| 5 | Air | 25–125 | 700 | 2 | 1:30 | 13.20 | 22.41 |
| 6 | Air | 25–125 | 850 | 3 | 1:20 | 4.88 | 13.77 |
| 7 | Air | Over 125 | 550 | 2 | 1:20 | 9.02 | 19.10 |
| 8 | Air | Over 125 | 700 | 3 | 1:25 | 10.40 | 20.34 |
| 9 | $N_2$ | Over 125 | 850 | 1 | 1:30 | 11.70 | 21.36 |
| 10 | $N_2$ | 0–25 | 550 | 3 | 1:30 | 64.03 | 36.13 |
| 11 | $N_2$ | 0–25 | 700 | 1 | 1:20 | 58.29 | 35.31 |
| 12 | $N_2$ | 0–25 | 850 | 2 | 1:25 | 17.33 | 24.78 |
| 13 | $N_2$ | 25–125 | 550 | 2 | 1:30 | 63.11 | 36.00 |
| 14 | $N_2$ | 25–125 | 700 | 3 | 1:20 | 61.23 | 35.74 |
| 15 | $N_2$ | 25–125 | 850 | 1 | 1:25 | 61.32 | 35.75 |
| 16 | $N_2$ | Over 125 | 550 | 3 | 1:25 | 61.48 | 35.77 |
| 17 | $N_2$ | Over 125 | 700 | 1 | 1:30 | 64.06 | 36.13 |
| 18 | $N_2$ | Over 125 | 850 | 2 | 1:20 | 54.39 | 34.71 |

**Table 5.** S/N response table for lithium leaching ratio.

| Symbol | Lithium Leaching Parameter | Mean S/N Ratio (dB) | | | |
|---|---|---|---|---|---|
| | | Level 1 | Level 2 | Level 3 | Max–min |
| A | Atomosphere | 22.18 | 34.48 | - | 12.3 |
| B | Powder size | 29.1 | 27.98 | 27.9 | 1.2 |
| C | Roasting temp. | 29.87 | 29.89 | 25.22 | 4.67 |
| D | Roasting time | 30.13 | 27.74 | 27.12 | 3.01 |
| E | Solid-liquid ratio | 27.78 | 28.38 | 28.83 | 1.05 |

*4.3. Analysis of Variance*

The ANOVA was conducted to investigate whether the design parameters have a significant impact on the experimental results. The total variance of S/N ratio, measured by the sum of squares of deviation from the total mean S/N ratios, is separated by the contribution of each design parameter and error to perform the calculation. First, the sum of the squared deviations from the total mean S/N ratio can be calculated as follows:

$$SS_T = \sum_{i=1}^{n} (\eta_i - \eta_m)^2 \tag{6}$$

In Equation (6), n denotes the number of experiments of OA, and $\eta_i$ represents the mean S/N ratio of the *i*th experiment. The impact of the design parameter on the characteristic is confirmed through the F test. When the F test is performed using the mean of the squared deviations for each design parameter, the mean of the squared deviations of each design parameter should be calculated. The mean of squared deviations (SSm) is equal to the sum of squared deviations (SSd) divided by the number of degrees of freedom associated with the design parameter. Then, the F-value for each design parameter is simply the ratio of SSm to the mean squared error. In general, the larger the F-value, the greater the impact on the characteristic.

Table 6 shows the impact on the Li leaching ratio based on the S/N ratio analysis and ANOVA. It can be seen that the heat treatment condition is the parameter that has the greatest impact on Li leaching.

**Table 6.** Results of the ANOVA for selective lithium leaching ratio.

| Symbol | Selective Lithium Leaching Parameter | Degress of Squares | Sum of Squares | Mean Square | F Test | Contribution (%) |
|---|---|---|---|---|---|---|
| A | Atomosphere | 1 | 645.06 | 645.06 | 40.86 | 72.28 |
| B | Powder size | 2 | 3.60 | 1.80 | 0.11 | 0.40 |
| C | Roasting Temp. | 2 | 84.19 | 42.10 | 2.67 | 9.43 |
| D | Roasting Time | 2 | 32.99 | 16.49 | 1.04 | 3.70 |
| E | Solid-liquid ratio | 2 | 0.29 | 0.14 | 0.01 | 0.03 |
| Error | - | 2 | 126.28 | 15.79 | - | 14.15 |
| Total | - | 11 | 892.41 | - | - | 100 |

*4.4. Confirmation Tests*

Once the optimal design parameter levels are selected, the last step is to predict and verify the improvement of the product quality characteristic by using the optimal design parameter levels. The results can be predicted using optimal parameter levels. The predicted values can be calculated as follows:

$$\hat{\eta} = \eta_m + \sum_{i=1}^{n}(\overline{\eta}_i - \eta_m) \tag{7}$$

In Equation (7), $\eta_m$ represents the total mean S/N ratio, $\eta_i$ represents the mean S/N ratio of the optimal level, and n is the number of major design parameters affecting the quality characteristic.

The estimated S/N ratio can be obtained using the optimal leaching parameters for selective Li leaching, and the Li leaching ratio under the experimental conditions can be calculated. Table 7 compares the actual leaching ratio with the Li leaching ratio predicted using the optimal Li leaching parameter levels, and identical leaching ratios are observed.

**Table 7.** Results of the confirmation experiment for lithium leaching ratio.

| A2B1C2D1E3 | Lithium Leaching Ratio (%) |
|---|---|
| Prediction | 65.73 |
| Experiment | 67.13 |

## 5. Analysis of the Effect of Heat Treatment on Lithium Leaching

The analysis results using the Taguchi method confirmed that the heat treatment condition had the greatest effect. Accordingly, the heat treatment was performed for 1 h by changing the heat treatment temperature in the nitrogen atmosphere; then, the leaching was performed in deionized water with an L/S ratio of 1:30. The results are shown in Figure 6a. When the DOE was performed using the Taguchi method, more detailed changes could be seen compared to those for the values experimentally designed with 550, 700, and 850 °C. Additionally, the highest Li leaching ratio of 70% was observed at 650 °C.

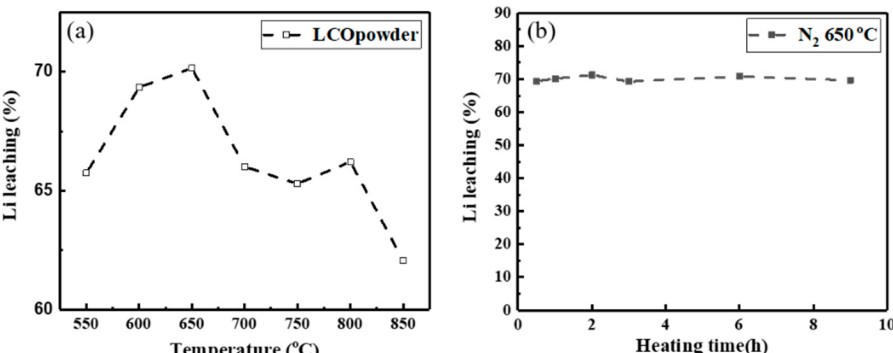

**Figure 6.** Leaching result after heat treatment by (**a**) temperature and (**b**) time in nitrogen atmosphere.

Subsequently, the experiment was conducted by changing the heat treatment time at 650 °C in a nitrogen atmosphere. The experimental results are shown in Figure 6b. When the heat treatment was performed for 2 h, a 70.5% Li leaching ratio was obtained, and no other metal leaching was observed. This is because Li is present in lithium carbonate and can be leached in deionized water, while leaching does not occur for other metals.

## 6. Results and Discussion

This study was conducted to optimize Li leaching in spent LIBs. An efficient Li recovery process was proposed, and the following results were obtained by carrying out the heat treatment and leaching process.

Optimal design using the Taguchi method provides a more systematic and efficient method than the majority of optimization techniques. The experiment was conducted 18 times with a total of four factors: the heat treatment condition (atmosphere/temperature and time), powder size, and L/S ratio. Subsequently, the analyses were performed for S/N ratio, ANOVA, and interactions to determine the effects on the quality characteristics and optimal values.

The S/N ratio is displayed in Figure 7. As shown in the graph, the heat treatment atmosphere had the greatest impact, followed by temperature, heat treatment time, powder size, and L/S ratio. For the optimal condition when the DOE was performed using the Taguchi method, 1 h heat treatment at 700 °C in nitrogen atmosphere was found to be optimal, and the Li leaching ratio was highest when leached in DI water at a 1:30 L/S ratio. The Li leaching ratio predicted using the parameter levels was 65.73%, and the actual leaching ratio was 67.13%, which was similar.

Chemical transformation is required for selective lithium leaching in black powder with carbon. A carbothermic reaction occurs at a high temperature. The heat treatment temperature at which black powder reacts is selected by TG-DSC analysis. Heat treatment is performed under the conditions of 550–850 °C in air/nitrogen atmosphere in order to know the chemical transformation of NCM. The phase was analyzed by XRD. As a result of XRD analysis, a $Li_2CO_3$ peak is observed at 700 °C. After the heat treatment at 850 °C, a peak of $LiAlO_2$ is confirmed because $Li_2CO_3$ is reacted with $Al_2O_3$ at a high temperature to form $LiAlO_2$.

The experiments were conducted for the heat treatment temperature and time in a nitrogen atmosphere for the detailed experiments on the parameters having large effects on the selective Li leaching. In the results, the optimal value was obtained at 650 °C, with Li leaching of over 70%. When heat treated at 650 °C, the effect of time on the Li leaching ratio was insignificant.

White powder was obtained through solid–liquid separation and evaporation concentration after performing the heat treatment under the optimal conditions. In the XRD analysis results, $Li_2CO_3$ powder was observed, as shown in Figure 8.

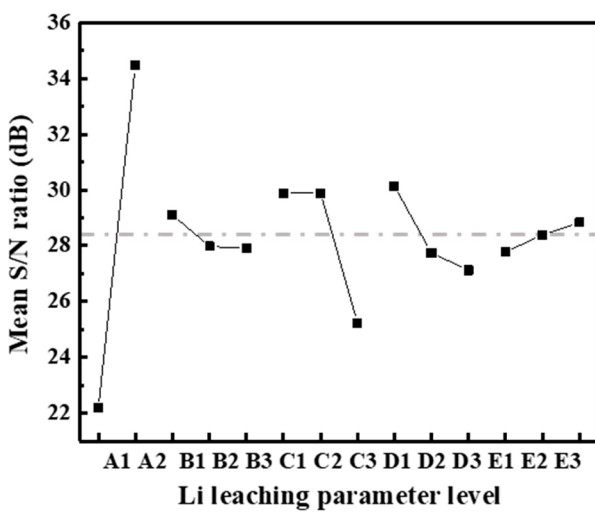

**Figure 7.** S/N graph for lithium leaching ratio.

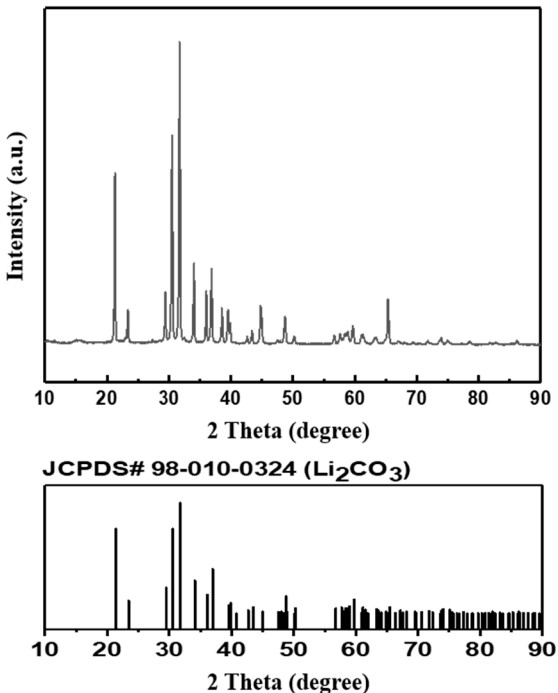

**Figure 8.** XRD analysis of evaporative concentration after solid–liquid separation.

**Author Contributions:** Conceptualization, Y.J.J., S.C.P., B.Y.Y., and S.H.S.; Data curation, Y.J.J.; Formal analysis, Y.J.J.; Investigation, S.C.P. and B.Y.Y.; Methodology, Y.J.J.; Project administration, S.H.S.; Supervision, S.H.S.; Validation, S.H.S. and B.Y.Y.; Visualization, S.C.P.; Writing—original draft, Y.J.J.; Writing—review and editing, S.H.S. All authors have read and agreed to the published version of the manuscript.

**Funding:** This study was supported by the Technology Innovation Program (Development of Material Component Technology) (Project No. 2011183) funded by the Ministry of Trade, Industry and Energy, Republic of Korea.

**Acknowledgments:** The authors are thankful for the support by the Ministry of Trade, Industry and Energy, Republic of Korea.

**Conflicts of Interest:** There is no conflict of interest to declare.

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
