# Peer review of "Design Optimization of Selective Lithium Leaching of Cathodic Active Materials from Spent Lithium-Ion Batteries Based on the Taguchi Method"

_metals, doi:10.3390/met11010108_

Round 1

Reviewer 1 Report

The manuscript presents an interesting topic. The recovery and reuse of metals is a topic of growing interest in the circular economy.

However, how the results are presented needs to be markedly improved. The introduction can be improved, and the main objective of the work must be emphasized. This is not clear.

The Results and discussion section is very poor.

More obtained results throughout the process should be shown.

Some points to improve:

-In Table 1 the authors indicate that the atmosphere parameters varied were N2 or air, the flow was modified? How the authors modify the powder size?

-The X-ray diffraction pattern of the initial black mass should be improved.

-Please, improve the quality of Tables. Modify them for great clarity for the reader.

The manuscript in the original form must be significantly improved to be accepted for publication.

Reviewer 2 Report

Manuscript reports a study volved to optimize Li leaching in spent lithium ion batteries (LIBs).

To analyze the effects of selective Li leaching the Taguchi method was used and an efficient Li recovery process was proposed

Study is well conducted with a detailed description of the procedure.  Results are interesting for their implication in recovery metals processes.

I think the manuscript can be accepted in this form and it needs no revision

Author Response

Thank you for reviewing our paper

Reviewer 3 Report

Dear Authors,

this paper contains interesting points, but it needs further improvement.

Li black powder and following leaching must have various reactions, but this paper contains only one reaction. possible reaction during heating and leaching should be noted. 

page 3, line 116, Table 3 seems Table 2.

Reference 18, LiCoO2 -> LiCoO2

after correction it must be good paper.

Round 2

Reviewer 1 Report

The manuscript has been slightly improved. It could be consider for publication.